# Revisiting Evaluation of Knowledge Base Completion Models

**Pouya Pezeshkpour**                                   PEZESHKP@UCI.EDU
*University of California, Irvine*

**Yifan Tian**                                            YIFANT@UCI.EDU
*University of California, Irvine*

**Sameer Singh**                                         SAMEER@UCI.EDU
*University of California, Irvine*

## Abstract

Representing knowledge graphs (KGs) by learning embeddings for entities and relations has led to accurate models for existing KG completion benchmarks. However, due to the open-world assumption of existing KGs, evaluation of KG completion uses ranking metrics and triple classification with negative samples, and is thus unable to directly assess models on the goals of the task: *completion*. In this paper, we first study the shortcomings of these evaluation metrics. Specifically, we demonstrate that these metrics (1) are unreliable for estimating how calibrated the models are, (2) make strong assumptions that are often violated, and 3) do not sufficiently, and consistently, differentiate embedding methods from each other, or from simpler approaches. To address these issues, we gather a semi-complete KG referred as YAGO3-TC, using a random subgraph from the test and validation data of YAGO3-10, which enables us to compute accurate triple classification accuracy on this data. Conducting thorough experiments on existing models, we provide new insights and directions for the KG completion research. Along with the dataset and the open source implementation of the models, we also provide a leaderboard for knowledge graph completion that consists of a hidden, and growing, test set, available at https://pouyapez.github.io/yago3-tc/.

## 1. Introduction

Knowledge graphs (KGs) are essential components of a wide range of tasks in scientific and industrial processes [Zhang et al., 2016, Zhu et al., 2018]. Most knowledge graphs, in practice, are often substantially incomplete and contain noise even in the edges they do have, prompting the need for models for knowledge graph completion (KGC), also called link prediction. In recent years, models that have led to accurate link prediction are based primarily on relational embeddings [Bordes et al., 2013a, Yang et al., 2015], where dense vectors are learned for each entity and relation in the KG. By using different scoring functions that capture the uncertainty in each fact [Trouillon et al., 2017, Dettmers et al., 2018, Sun et al., 2019a], such knowledge graph completion models have achieved incredible success on existing benchmarks.

Unfortunately, the lack of a complete and accurate KGs is a problem for evaluation as well. Since it is not possible to list *all possible* true and false facts for a KG of interest, existing evaluation of KGC consists of gathering known *true* facts, and using: (1) *ranking metrics*, such as Hits@N and Mean Reciprocal Rank (MRR), to calculate the relative rank of these known true facts against all unknown facts (thus implicitly treated as negative), and (2) *classification accuracy* of individual facts, by treating random corruptions of a known true fact as negative/false facts. In spite of steady and significant progress on these models, it is not clear whether these metrics correspond to the true performance on link prediction, making it difficult to decide whether they are ready for real-world

deployment. Further, due to the strong assumptions made by these evaluation metrics, the strengths, shortcomings, and reasoning capabilities underlying these link prediction methods is difficult to determine, hindering further progress of the field.

In this paper, we study significant issues with the current evaluation metrics for knowledge graph completion models, in particular, highlighting the impact of the assumptions made by these metrics on model performance. We show that the ranking metrics often do not correlate well with the actual performance of the model, make it incredibly challenging to determine whether these models are well-calibrated or not (an essential property for real-world deployment), and do not correlate well with the reasoning power of the models. For triple classification, upon a detailed examination of several commonly used benchmarks, we show that the metric is heavily sensitive to the choice of negative sampling, and that there is a significant mismatch between accuracy and the ranking metrics.

To address these shortcomings in existing benchmarks, we introduce **YAGO3-TC**, a high-quality, manually-annotated dense sub-graph of the YAGO3-10 KG. Along with the true facts that are already present in test and validation splits of the existing benchmark, YAGO3-TC also includes related facts involving the same entities that are annotated to be true or false via crowdsourcing. These related facts are designed to be somewhat challenging to discriminate since they are high-scoring by recent accurate models, resulting in 28,364 labeled facts out of which 2,976 are positive. Since we ensure the quality of the annotations, classification metrics such as accuracy, precision/recall, etc. can be used to appropriately evaluate models of knowledge graph completion.

We also provide a comprehensive analysis of recent KG completion models, given the high-quality annotations in YAGO3-TC, using triple classification metrics. We are able to provide accurate calibration results for completion models, showing that they are significantly overconfident (consistent with existing results for neural networks, but different from other observations for KGC). Further, we observe that there is a significant mismatch between ranking metrics and performance on the completion task (e.g., there is more than 20% gap between Hits@1 and Precision). Most importantly, we show that the progress in performance indicated by ranking metrics does not align with actual completion task; simple methods achieve similar performance to state-of-art models.

## 2. Background and Notation

In this section, we introduce notations, benchmarks, evaluation procedures, and a brief overview of existing relational embedding approaches to knowledge graph completion. More details on existing benchmarks and implementation details are provided in the Appendix.

**Embedding Based KGC:** To represent the facts in the KGs, we use triples of subject, relation, and object, $\langle s, r, o \rangle$, where $s, o \in \xi$, the set of entities, and $r \in \mathbb{R}$, the set of relations. To model the KG for link prediction, a scoring function $\psi : \xi \times \mathbb{R} \times \xi \to \mathbb{R}$ is learned to evaluate whether any given fact is true. In this work, we will primarily study DistMult [Yang et al., 2015] due to its simplicity and popularity, and RotatE [Sun et al., 2019a] and Tucker [Balazevic et al., 2019] because of their state-of-the-art performance. Only true facts are included in the training data, thus training the model involves, for each observed triple, sampling $L$ negative ones (value of $L$ is usually treated as a hyperparameter) by randomly corrupting either the subject or the object of the triple. Using the model's probability of truth as $\sigma(\psi(s, r, o))$ for $\langle s, r, o \rangle$, following Yang et al. [2015], Trouillon et al.

[2016], the binary cross-entropy loss is defined as:

$$\mathcal{L}(G) = \sum_{(s,r)\in G} \sum_{o} y_o^{s,r} \log(\sigma(\psi(s,r,o))) + (1 - y_o^{s,r}) \log(1 - \sigma(\psi(s,r,o))). \qquad (1)$$

where $y_o^{s,r}$ represents whther the fact is "true" ($y_o^{s,r} = 1$ for observed facts, $y_o^{s,r} = 0$ otherwise).

**Ranking Metrics:** To evaluate the performance of the KG completion models, we rank test triples against all possible negative samples, generated by corrupting the subject or object of the target triple. Ranking metrics have been used since existing KGs are open-world and the ground truth label for all negative and positive samples is not available. In the *filtered* setting, which we consider in this work, we only treat triples that do not appear in the training, validation, or test set, as the possible negative samples. To quantify the ranking of target triples, we use standard evaluation metrics such as Mean Reciprocal Rank (MRR), which is the average inverse rank of all test triples, and Hits@N, which is the percentage of test triples whose rank is lower (better) than or equal to N.

**Triple classification:** Triple classification is the task of binary classification on the KGs triples. This task is important because, if appropriately set up, it directly evaluates the capability of KGC models in identifying missing links. Specifically, given a target triple $\langle s, r, o \rangle$, we want to identify if this is a positive/true fact or a negative one. For this task, previous approaches learn a specific threshold $\tau_r$ for each relation, over validation data. In order to create the negative samples in these approaches, for both validation and test data, they corrupt subject or object of the target triple with a random entity form the KG. After learning thresholds $\tau_r$, a triple assumed to be positive if its score is higher than the threshold for the triple's relation.

## 3. Issues in Existing KG Completion Evaluation

In this section, we discuss some issues prevalent in current evaluation metrics, and provide empirical evidence for their shortcomings. First, we observe that the assumptions underlying ranking metrics are often incompatible with the goals of the completion itself. Then, we show that evaluating how well completion models are calibrated is challenging since the results are incredibly sensitive to the setup design choices. Finally, we show that the results of the ranking metrics are often inconsistent with the results of the triple classification evaluation.

### 3.1 Assumptions in Ranking Metrics

In the past few years, we have observed tremendous progress in the performance of KG completion models, based on ranking metrics. As these models become increasingly accurate and potentially ready for real-world deployment, it is now useful to understand the extent to which these ranking metrics align with the actual goals of the *completion* task.

Let us consider a simple example. Assume we want to validate whether triple $\langle s, r, o \rangle$ is true or not. According to the current procedure, our only option is to rank the score of all possible objects (triples of the form $\langle s, r, o' \rangle$) and subjects (of the form $\langle s', r, o \rangle$) and compute the rank of our target triple. In this case, the ranking metrics such as Hits@N can only tell us whether this triple appears in the top N possible triples. If the relation of our target triple can accept only one true object, i.e. the relation is N-1, this ranking is meaningful, however, if multiple objects can be true for our target subject and relation, this ranking is incomplete since does not capture other triples that are ranked higher than the target fact are themselves true or not. A similar observation holds for relations for

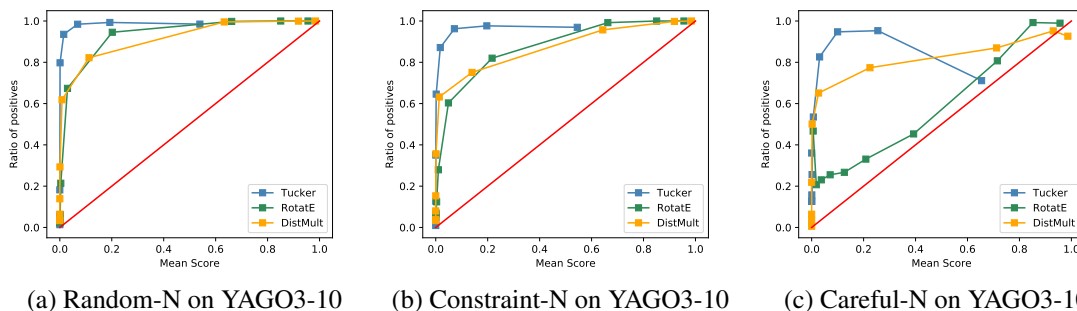

(a) Random-N on YAGO3-10     (b) Constraint-N on YAGO3-10     (c) Careful-N on YAGO3-10

Figure 1: Calibration study on different KGs based on three negative sampling procedures. We plot reliability diagrams of the fraction of positive triples to all the triples vs the link prediction models' score for a target triple. Being closer to the diagonal means the model is more calibrated.

which multiple subjects can be true for the same object, i.e. a 1-N relation. Unfortunately, on studying two commonly used KGs WN18RR and YAGO3-10, we notice that this phenomenon happens in a huge portion of the data, as a result of the existence of semi-inverse relations.

**Semi-Inverse Relations in WN18RR:** On conducting simple statistical analysis on WN18RR, we notice that this KG is not completely free of relations that are inverse of each other (which make the completion task trivial). We notice in more than $90\%$ appearance of three relations *_derivationally_related_form*, *_verb_group*, and *_similar_to*, and also more than $60\%$ appearance of *_also_see*, another triple with the same entities but the opposite direction of the original relation appears in the training data. Together, these relations consist $37\%$ of this KG. Moreover, around one-third of triples in the test data contain one of these relations; the KGC models achieve more than $0.95$ MRR performance on this subset, significantly affecting the overall performance.

**Semi-Inverse Relations in YAGO3-10:** Along the similar lines, in YAGO3-10, for $75\%$ of the triples with relation *isAffiliatedTo*, the triple with relation *playsFor* appears between the same object and subject ($87\%$ for reverse). These relations consist of $63\%$ of test data, $81\%$ of which have the other relation between the same subject and object in the training data. Note that embedding methods achieve around $0.95$ of MRR performance on these triples.

The existence of these semi-inverse relations results in two important conclusions. First, it indicates that the performance based on ranking metrics on these KGs is not trustworthy since it is very easy to predict these triples. Second, since these relations can accept many objects (and subjects) for the same subject (and object), and embedding methods score all of those objects (and subject) *very highly* (since their semi-inverse version of them appear in the training data), low values of ranking metrics are clearly not accurate assessment of completion on these triples. More specifically, for a target triple with one of the mentioned relations, as the number of *true* objects or subjects with semi-inverse relation increases, the chance of obtaining a worse ranking increases as well.

### 3.2 Evaluating Calibration of the Models

Calibration is a very important aspect of KG completion that has only recently received attention [Tabacof and Costabello, 2019]. Treating the probability of truth of a fact ($\sigma(\psi(s,r,o))$) for triple $\langle s, r, o \rangle$) as the confidence of the model for the triple, we consider our model to be calibrated if the confidence aligns with the rate of true facts. In other words, if confidence is equal to $0.5$, we

expect to have around $50\%$ of triples with this confidence to be true. If this proportion is far from $50\%$, then the model is not calibrated (the model is under-confident if the proportion is higher, and over-confident if it is lower). Since the evaluation only consists of true facts, we need to obtain negative/false facts by sampling. We use three different negative sampling procedures: (1) randomly replacing subject or object with an entity from all possible ones (*Random-N*), commonly used in KG completion literature, (2) randomly replacing subject or object with an entity that has appeared at least once with the target relation in the training data (*Constraint-N*), which was used by Socher et al. [2013] to generate more challenging negative samples, and (3) choose the highest scoring negative sample that has the object (or subject) with a different type than the target triple object (*Careful-N*). By choosing the object (or subject) that has a different type than the target triple entities we enforce the chosen negative sample be a true negative. We define entity type as the set of entities that have appeared with similar relations (see appendix for a precise definition).

The result of calibration study based on above negative samples is shown for YAGO3-10 in Figure 1 (calibration plot for WN18RR and FB15k-237 and histogram plot of score distributions is provided in the appendix). Note that even though negative samples for these three negative sampling methods are different, we generate each plot on the same set of negative samples for the three models. Although these results show that RotatE provides more calibrated models compared to Tucker in all the negative sampling procedures, the advantage of RotatE over DistMult changes with different negative samples (in *Random-N*, DistMult appears better than RotatE). Further, we suspect the reason behind the peculiar behavior of Tucker in Figure 1c is due to the fact that Tucker tends to score many triples (both positive and negative triples) very highly for specific relations, such as *hasGender* and *isLocatedIn*. Moreover, as we make the negative sampling more challenging, we see extremely different behavior from the models, some result in a much more calibrated plot compared to others (e.g. RotatE looks calibrated for *Careful-N*, but not for the rest), making these benchmarks inconclusive for calibration. For WN18RR and FB15k-237, although DistMult outperforms the other two methods completely, we observe similar behaviors in calibration plots. Last, these plots also indicate that the models are under-confident, which is inconsistent with similar studies on neural networks [Guo et al., 2017]. For a comparison that takes the model complexity into account, we include results for models that have the same number of parameters in the appendix.

### 3.3 Simple Models Look Accurate

In this section, we evaluate the reasoning capabilities of current link prediction methods. More specifically, we wanted to see how far in performance on ranking metrics we can get to by adopting very simple methods and see if ranking metrics can properly differentiate between SOTA models and these simple approaches. We first study rule-based methods that only predict ranking for triples that have their semi-inverse relations in the training data. Then, introducing a *local score* that learns simple neighborhood patterns, we see unexpectedly high performance on ranking metrics, casting doubt on the capability of ranking metrics to accurately evaluate KGC methods.

**Rule-based Link Prediction:** To see the effect of semi-inverse relations on the performance of link prediction methods, we provide a very simple rule-based method. For WN18RR and YAGO3-10 target triples, we identify all objects for which the target subject appears with a semi-inverse relation in the training data, and vice versa for the subjects. Then we rank these entities based on their popularity (their degree in the graph) in the KG. The result of this rule-based method is provided in Table 1. As shown, for both of YAGO3-10 and WN18RR, this method achieves high performance.

Table 1: **Link Prediction** result for FB15k-237, WN18RR and YAGO3-10 KGs. All results generated using perspective models' SOTA hyperparameters.

| Models | FB15k-237 | | | WN18RR | | | YAGO3-10 | | |
|---|---|---|---|---|---|---|---|---|---|
| | MRR | Hits@1 | # Param | MRR | Hits@1 | # Param | MRR | Hits@1 | # Param |
| DistMult | 0.295 | 19.8 | 5.8M | 0.428 | 39.2 | 8.1M | 0.409 | 31.2 | 24.6M |
| RotatE | 0.331 | 23.4 | 29.3M | 0.478 | 43.4 | 40.9M | 0.471 | 38.2 | 123.2M |
| Tucker | 0.342 | 25.1 | 11M | 0.456 | 42.8 | 9.4M | 0.468 | 37.9 | 63.9M |
| Rule-Based | - | - | - | 0.338 | 32.1 | - | 0.286 | 24.2 | - |
| Local | 0.181 | 12.7 | 0.3M | 0.364 | 33.4 | 2k | 0.322 | 25.8 | 50k |

**Local Score:** We also study an alternate simple model, using just the local structure around the target triple. For each target triple, we compute a *local score* by finding all paths from the subject to object and score them in the context of the relation of target triple (a simpler version of this local score is studied in Toutanova and Chen [2015]). Specifically, we define the local score as: $\mathcal{L}oc(s, r, o) = \sigma(\sum_{p \in P(s,o)} W_r^p)$ , where $P(s, o)$ denotes the set of all the paths between $s$ and $o$. To learn $W_r^p$ we generate negative samples by randomly corrupting the $r$. A visual representation of local score is provided in the appendix. For FB15K-237 that is denser than WN18RR and YAGO3-10, for each relation, we consider the top 5 most frequent paths with length 2 between the subject and the object of triples with that relation[1]. For WN18RR and YAGO3-10, since most of the triples do not have paths with length 2 between their entities, we only score simple patterns with length 3 in our model. More specifically, these paths comprise of patterns that have one edge with the same relation as the target sample (with the same direction). Further, they should have the same relation for the other two edges but in a different direction. More details and visualization of these patterns are provided in the appendix. The result of link prediction on our benchmarks is provided in Table 1. As it shows, in all three KGs, our local score performs comparably to embedding methods while having a much fewer number of parameters. The high performance of both these simple models raises questions about the utility of ranking metrics and existing benchmarks.

### 3.4 Problems with Triple Classification with Negative Sampling

To demonstrate that the current approach to evaluating accuracy via triple classification is inaccurate, we create a fact classifier from completion models (that only provide a score) by learning a threshold for each relation (following standard practice). The performance of state-of-art embedding models over several KGs is provided in Table 2. As it shows, all the models achieve very high accuracy performance (around $80\%$) when we choose both validation and test negative samples randomly (*Random-N*). The reason behind this high performance is mostly due to the naive way of random negative sampling. Moreover, Distmult and RotatE outperform the Tucker performance in all three KG (although Tucker often achieves higher or very similar performance on ranking metrics).

**Negative Sampling:** There are a few fundamental issues with the current approach to triple classification: (1) randomly choosing the negative samples results in very simple classification, which is not informative for evaluation, and (2) training classifiers (estimating thresholds) based on random negative samples makes the results brittle, i.e., choosing slightly more challenging negative samples

---
1. Around $70\%$ the test triples have at least one of these patterns in their neighborhood

Table 2: Triple classification accuracy for random and careful negative sampling.

| Models | FB15k-237 | | WN18RR | | YAGO3-10 | |
|---|---|---|---|---|---|---|
| | Random-N | Careful-N | Random-N | Careful-N | Random-N | Careful-N |
| DistMult | 95.2 | 47.6 | 83.3 | 39.5 | 94.9 | 45.5 |
| RotatE | 94.4 | 49.1 | 84.8 | 42.0 | 86.1 | 42.9 |
| Tucker | 77.6 | 57.4 | 72.0 | 55.8 | 75.4 | 45.2 |

Table 3: Triple classification accuracy on ground truth labels. The results are averaged over 5 runs.

| Models | Kinship | | | | Nations | | | |
|---|---|---|---|---|---|---|---|---|
| | Acc | F1 | Recall | Precision | Acc | F1 | Recall | Precision |
| Distmult | 58.8 | 7.6 | 34.8 | 4.3 | 86 | 24 | 25.6 | 22.9 |
| RotatE | 10.6 | 10 | 97.3 | 5.3 | 66.9 | 27.2 | 69.6 | 16.9 |
| Tucker | 86.2 | 38.8 | 83.7 | 25.2 | 55.6 | 18.9 | 66.6 | 11 |
| Type Constraint | 28.9 | 12.0 | 94.6 | 6.4 | 47.6 | 22.8 | 87.0 | 13.1 |

can reduce the performance dramatically. Instead of random samples, we instead use the challenging *Careful-N* samples described in Section 3.2, and show results in Table 2. As it shows, these negative samples dramatically reduce the accuracy. Further, although Tucker performs the worst in the random setting, here we see a smaller reduction in accuracy compared to the other two methods (RotatE appears better than DistMult). We suspect the reason behind this smaller reduction in accuracy is that Tucker distinguishes between positive and hard negative samples better, i.e., on average, assigns considerably higher scores to positive samples in comparison to hard negatives.

**Mismatch between Ranking Metrics and Accuracy:** We also evaluate these models on triple classification in Table 3 for Kinship and Nations, which have *all* the true facts available (all missing facts are false, and can be enumerated). As the negative samples, we consider the union of top 10 negative objects and subjects based on trained RotatE and Tucker models. Although these models achieve around 0.8 MRR and 100% Hits@10 performance, they demonstrate much lower performance on accuracy metrics showing that these ranking metrics are not trustworthy. Moreover, if we classify the negative and positive samples for Kinship and Nations KGs only based on the compatibility of their subject and object with the relation, based on our defined notion of type (*Type Constraint*), we see that *Type Constraint* achieves comparable results with these embedding methods, questioning the credibility of the ranking metrics and performance of these embedding methods.

## 4. YAGO3-TC: A New Benchmark for Evaluating KG Completion

In this section, we first describe our procedure to gather YAGO3-TC dataset, and then, we explain our plans to continuously update YAGO3-TC as new KG completion models are proposed.

### 4.1 Creating YAGO3-TC

To solve these issues with KG completion evaluation, we gather a dataset that contains true and false facts, but is also challenging for current models. Note that in this work, we are not suggesting that

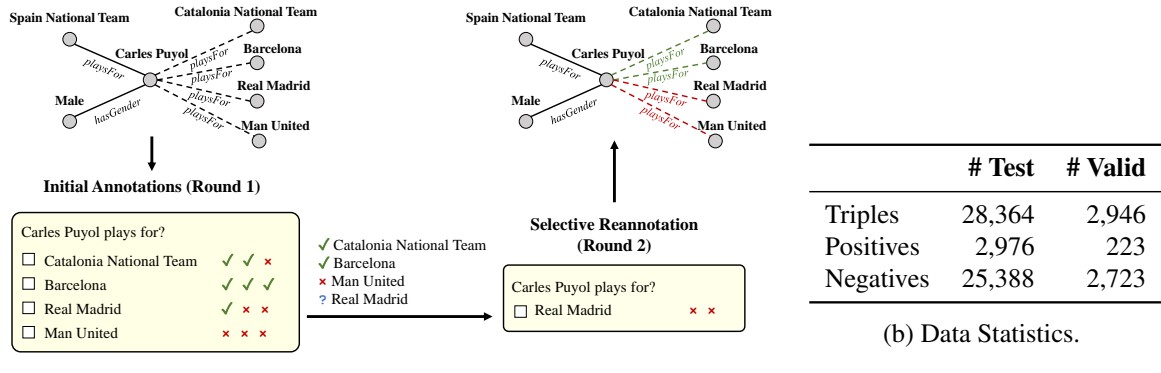

(a) Overview of crowdsourcing process.

| | # Test | # Valid |
|---|---|---|
| Triples | 28,364 | 2,946 |
| Positives | 2,976 | 223 |
| Negatives | 25,388 | 2,723 |

(b) Data Statistics.

Figure 2: **YAGO3-TC Dataset.** (a) annotation process, and (b) statistics of the resulting data

we should completely replace the ranking metrics for all use cases, but point out their shortcomings, and introduece a benchmark to compute other metrics. Since each embedding model scores triples differently, we use RotatE [Sun et al., 2019a] and Tucker [Balazevic et al., 2019] as our judges for identifying important triples. More specifically, we first sample 1000 random triples from the test set of YAGO3-10 (the relation distribution histogram of YAGO3-10 test data and our 1000 sampled triples is very similar, provided in the appendix). To reduce the effect of semi-inverse relations, we do not consider triples with relation *isAffiliatedTo* in our samples. Then, applying trained RotatE and Tucker on these triples we find the 10 top scoring objects for the query $\langle s, r, ? \rangle$ and 10 top scoring subjects for $\langle ?, r, o \rangle$. Excluding repeated triples, we gather 28,364 triples/facts.

We need to label these triples as negative (false) and positive (true). Before conducting crowd-sourcing to label these triples, there are few criteria to identify the true label of some of these triples: (1) if the relation of target triple is N-1 (or 1-N) we can treat every object (or subject) as a negative sample, except for the original target object (or subject), and (2) if the object (subject) of a sample does not have the same type as the object (subject) of the original test triple, we can treat that sample as negative. Filtering these identifiable samples, we label the rest through crowd-sourcing.

For labeling the samples, we ask the users to search the information on the Wikipedia page of the entities and use the Google search engine. For example, we ask users to choose all the correct teams for the query "Carles Puyol plays for?" from provided options. We use Amazon Mechanical Turk as the framework for gathering the labels and ask three users to answer each query. If more than one user agree on an answer, we treat that triple as a positive one. We separately reannotate the objects that were picked only by one of the users. This time, we ask two users to check these samples, and if both of them agree on a correctness of a choice, we treat it as a positive sample. After creating the set of positive labels, we treat everything else as negative samples. An overview of our user study is depicted in Figure 2a. Further, after randomly choosing 100 samples from the validation data of YAGO3-10, we use the same procedure for gathering labels for our validation data. Since we intend to use these labels to find the thresholds of the models, we ensure that at least one triple for each relation that appears in this set. The dataset statistics are provided in Table 2b. To check the quality of our labels, we also include 100 true facts from the original test data in our study, and find that 96% of these triples were annotated to be positive, demonstrating the high quality of our labels.

| Models | Acc | F1 | R | P | A-ROC |
|---|---|---|---|---|---|
| DistMult | 29.4 | 20.4 | **86.6** | 11.6 | 0.61 |
| RotatE | 27.0 | 19.4 | 83.7 | 10.9 | 0.58 |
| Tucker | 63.3 | **22.3** | 50.3 | 14.4 | **0.64** |
| DistMult-valid | 85.6 | 19.1 | 14.3 | 29.1 | 0.59 |
| RotatE-valid | **88.6** | 18.9 | 12.8 | **42.1** | 0.61 |
| Tucker-valid | 79.7 | 22.1 | 27.5 | 18.5 | 0.56 |
| Random | 80.9 | 10.9 | 11.1 | 10.6 | 0.51 |
| Type Constraint | 32.2 | 20.8 | 84.8 | 11.8 | 0.61 |
| Local | 61.0 | 19.0 | 43.8 | 12.2 | 0.6 |

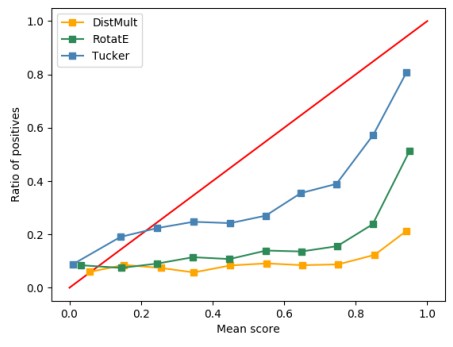

(b) Calibration plot for YAGO3-TC.

(a) Triple classification accuracy on ground truth labels. The results are averaged over 5 runs.

Figure 3: **Triple classification on YAGO3-TC.** (a) provides average performance of embedding methods and our baselines. (b) Depicts the calibration study of embedding models.

## 4.2 Continuously Updated, Hidden Benchmark

Although we are confident about the quality of the true/false annotations, we select the candidates based on the output of the two recent models, RotatE and Tucker. As new models will be proposed, they may be able to differentiate between our gathered true and false facts, but may highly rank facts that are not in our dataset. In order to maintain a benchmark that is useful for evaluating knowledge base completion in the long run, we propose a web-hosted evaluation platform. The online platform, available at https://pouyapez.github.io/yago3-tc/, includes a *hidden* test set, and a leaderboard of model submissions. The test set, initialized with the YAGO3-TC dataset described here, will continuously be updated as we receive model predictions from the submissions, thus identifying *more challenging* candidate facts, via the same crowdsourcing pipeline described above.

## 5. Evaluation Using YAGO3-TC

In this section, we investigate the triple classification evaluation using our new dataset. We first study the performance of several embedding models on the dataset and introduce new simple techniques to improve current models. Then, comparing the calibration of the triple classification task with the ranking scenario, we demonstrate this task is better defined. Finally, we study the per-relation breakdown of accuracy to better assess model performance.

### 5.1 Performance of Existing KGC Models on YAGO3-TC

To provide a better evaluation procedure for KG completion we study accuracy metrics on our gathered data. The averaged result of SOTA methods in YAGO3-TC over 5 runs is provided in Table 3a. As it shows, except for recall, Tucker outperforms RotatE. We note that, in this experiment, accuracy and precision are the metrics we care about the most because it is important to avoid labeling a triple as positive incorrectly. Moreover, comparing the results with ranking metrics in Table 1, we see a significant mismatch with these metrics and the ranking ones.

We also consider 3 baselines for our benchmark, 1) randomly assigning labels using the actual ratio of positives and negatives as the probability, 2) classifying based on compatibility of the type of subject and the object with the relation, and 3) train our classifier using the scores of our local models from Section 3.3. All of these baselines achieve comparable results with embedding methods demonstrating the need for better training and models. We are also interested in investigating whether we can improve these performances with simple modification on the learning process. First, instead of random negative sampling on validation data, use our gathered validation data as the training data for triple classification (*model-valid*). As shown, upon training on our validation data, the accuracy and precision increase dramatically and recall drops with a huge gap. To provide a deeper understanding of the performance, a per-relation breakdown is provided in the appendix.

## 5.2 Calibration

As we showed, calibration study on existing evaluation metrics is not a well-defined task. YAGO3-TC provides us with an opportunity to study calibration in a more controlled and representative environment. The evaluation on the calibration of YAGO3-TC is depicted in Figure 3b, with the histogram plot of the scores in the appendix. As shown, Tucker provides a more calibrated plot comparing to RotatE. Moreover, previous calibration curves suggested models are under-confident, whereas here the calibration reveals that they are overconfident, which is consistent with calibration studies on neural network models [Guo et al., 2017].

## 6. Related Work

There is a rich literature on representing knowledge bases using fixed-size embedding vectors.

In the past few years, a number of recent techniques have proposed models that firstly assign an embedding for each entity and relation, and then use these embeddings to predict facts. These methods which primarily only differ in scoring function for link prediction task, include tensor multiplication [Nickel et al., 2011, Socher et al., 2013, Yang et al., 2015, Balazevic et al., 2019], algebraic operations [Bordes et al., 2011, 2013b, Dasgupta et al., 2018, Sun et al., 2019a], and complex neural models [Dettmers et al., 2018, Nguyen et al., 2018]. Furthermore, a number of studies have examined the incorporation of extra types of evidence to achieve more informative embeddings, with extra modalities consisting of numerical values [Garcia-Duran and Niepert, 2017], images [Oñoro-Rubio et al., 2017], text [Toutanova et al., 2015, 2016, Tu et al., 2017], and their combinations [Pezeshkpour et al., 2018]. Utilizing the analysis in this work, we hope to shed more lights on better integrating extra modalities in the vector space to have more informed embeddings.

Although these methods provide accurate models for a variety of KG tasks, only a few try to provide a better understanding for these models, such as by addressing issues in training [Kadlec et al., 2017, Jain et al., 2020, Ruffinelli et al., 2020], investigating particular triples in the data [Akrami et al., 2020], studying sparsity and unreliability of KGs [Pujara et al., 2017], analyzing interpretability in the embedding space [Sharma et al., 2018, Pezeshkpour et al., 2019, Allen et al., 2019], and identifying existing issues in KG completion models [Sun et al., 2019b]. Although Tabacof and Costabello [2019] also study calibration of link prediction models, there are several differences between our study on calibration: 1) we show the effect of different negative sampling procedures on the calibration of link prediction methods, and 2) we further provide a well-defined and explicit environment for calibration study using our proposed YAGO3-TC. Moreover, Safavi et al. [2020] studies utility of KG embedding methods in real-world completion tasks by proposing

to calibrate these embedding models to output reliable confidence estimates for predicted triples. It is worth mentioning that developing more appropriate and challenging datasets as a way to address shortcoming of existing benchmarks has been used in other machine learning tasks, such as visual reasoning [Johnson et al., 2017, Kottur et al., 2019], semantic parsing [Nangia and Bowman, 2018] and textual entailment [Zellers et al., 2018], amongst others.

## 7. Conclusion

In this work, we set to investigate whether ranking metrics are appropriate measures to evaluate link prediction models. Upon, studying shortcoming and strength of the current adopted procedure, we first show existing issues with ranking metrics: they do not evaluate completion, are difficult to use for calibration, and are not able to consistently differentiate between different models. Facing these issues, after redefining the triple classification task, we gather a new dataset YAGO3-TC consisting of a dense subgraph annotated with both true and false facts. Exploring several SOTA embedding models on this dataset we further provide insights and directions for future works. We hope that this research and dataset will bridge the gap in better adoption of link prediction models in real-world scenarios. The datasets, leaderboard with continuously updated benchmark, and the open-source implementation of the models are available at https://pouyapez.github.io/yago3-tc/. We hope this annotation methodology is used for existing, and future, evaluation bechmarks in KG completion.

## Acknowledgements

We would like to thank Matt Gardner and the anonymous reviewers for their feedback. This work is supported in part by the DARPA MCS program under Contract No. N660011924033 with the United States Office of Naval Research, and in part by NSF award #IIS-1817183. The views expressed are those of the authors and do not reflect the official policy or position of the funding agencies.

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

## Appendix A. Scoring Functions and Implementation Details

Here we first describe different scoring functions adopted in this work and then elaborate the implementation details.

**Scoring Functions:** In DistMult, $\psi(s, r, o) = \mathbf{e}_s \mathbf{R}_r \mathbf{e}_o$, where $\mathbf{e}_s, \mathbf{e}_o \in \mathbb{R}^d$ are embeddings of the subject, and object and $\mathbf{R}_r \in \mathbb{R}^{d \times d}$ is a diagonal matrix representing the relation $r$. Moreover, The RotatE scoring function is defined as $\psi(s, r, o) = \parallel \mathbf{e}_s \circ \mathbf{R}_r - \mathbf{e}_o \parallel^2$ where $\mathbf{e}_s, \mathbf{R}_r, \mathbf{e}_o \in \mathbb{C}^d$ and $\circ$ denotes the Hadamard product. In Tucker, the score of triple $\langle s, r, o \rangle$ is defined as $\psi(s, r, o) = W \times_1 \mathbf{e}_s \times_2 \mathbf{R}_r \times_3 \mathbf{e}_o$, where $\mathbf{e}_s, \mathbf{e}_o \in \mathbb{R}^{d_e}$, $\mathbf{R}_r \in \mathbb{R}^{d_r}$, $W \in \mathbb{R}^{d_e, d_r, d_e}$ and $\times_i$ is representing the tensor product along the ith mode.

**Implementation Details:** We use the same loss and optimization for training, i.e., AdaGrad and the binary cross-entropy loss. We adopt reported hyperparameters from previous works to reproduce their performance. To investigate the link prediction task, we study commonly-used metrics for evaluation in this task: mean reciprocal rank (MRR) and Hits@N. As our embedding methods, we consider DistMult [Yang et al., 2015] because of its simplicity and high performance, and RotatE [Sun et al., 2019a] and Tucker [Balazevic et al., 2019] because of their state of the art performance. Further, we use validation data to tune the hyperparameters and use a grid search to find the best hyperparameters, such as regularization paramete. To evaluate our method, we conduct link prediction experiment on two small KGs Kinship and Nations and three more realistic KGs FB15k-237 [Toutanova et al., 2015], WN18-RR [Dettmers et al., 2018] and YAGO3-10 [Mahdisoltani et al., 2013]. A statistical analysis of our benchmarks is provided in Table 4

## Appendix B. Entity Types

**Definition B.1.** *In this work, we define a generic notion of type for entities. We consider two entities to have the same type if they appear with relations in the training data, that themselves have appeared several times with the same objects (subjects). More specifically, for target triple $\langle s, r, o \rangle$, to find all the entities with the same type as $s$, we first find all the relations that for some number of times, appear with the same entities for their subject as the relation $r$. Then we consider the union of all entities that appear as the subject for those relations in the training data, as the set of the same type entities for $s$. Throughout the paper, we use this notion of type to identify the type of each entity.*

Table 4: **Data Statistics** of the benchmarks.

|           | # Rel | #Ent    | # Training | #Test  | #Valid |
|-----------|-------|---------|------------|--------|--------|
| WN18RR    | 18    | 40,768  | 86,835     | 3,134  | 3,034  |
| FB15k-237 | 237   | 14,541  | 272,115    | 20,466 | 17,535 |
| YAGO3-10  | 37    | 123,170 | 1,079,040  | 5,000  | 5,000  |
| Nations   | 56    | 14      | 1,592      | 200    | 200    |
| Kinship   | 26    | 104     | 8,544      | 1,074  | 1,068  |

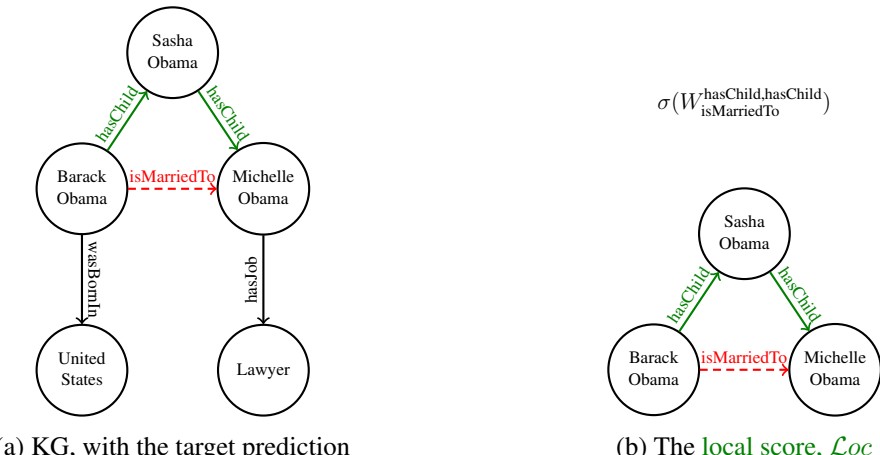

(a) KG, with the target prediction        (b) The local score, $\mathcal{L}oc$

Figure 4: Score of each triple includes local score, which captures paths between subject and object entity in the target triple.

## Appendix C. Local Score

In this section, we analysis the scoring function and the simple patterns that we incorporate to our model. A simple representation of our local model is depicted in Figure 4. Moreover, the simple patterns with length 3 that we consider for WN18RR and YAGO3-10 is depicted in Figure 5. The reason for choosing these patterns is the fact that they are very easy to learn. To learn these patterns, the translation based embedding method such as RotatE just need to learn that if a path contains two edges with the same relation but in the reverse direction, these edges would cancel each other out. And this is a direct result of the definition of translation based scoring function. For Multiplicative based embedding such as DistMult if we assume that $|\mathbf{e}_o|, |\mathbf{e}_s|, |\mathbf{e}_s\mathbf{R}_r| = 1$, the scoring function can be considered as translation-based embedding by considering the space angle as the metric of similarity instead of Euclidean distance.

## Appendix D. Calibration Study

The calibration plot for WN18RR and FB15k-237 over our three defined negative sampling procedure is depicted in Figure 6 The histogram plot of scores' distribution for WN18RR, FB15k-237 and YAGO3-10 using Distmult, Tucker and Rotate as link prediction models and adopting studied mentioned negative sampling procedures is depicted in Figure 7. Moreover, the histogram plot of scores' distribution for YAGO3-TC is depicted in Figure 8.

## Appendix E. Number of Parameters and Calibration

In this section, we reproduce the calibration plots by fixing the number of parameters over different models. We consider the DistMult's number of parameters with a hidden dimension of 200 as our benchmark. The MRR performance of different models with the same number of parameters is provided in Table 5. Moreover, the calibration plot using these models is depicted in Figure 9. As it shows, the results appear very similar to previously reported ones. The reason behind similar

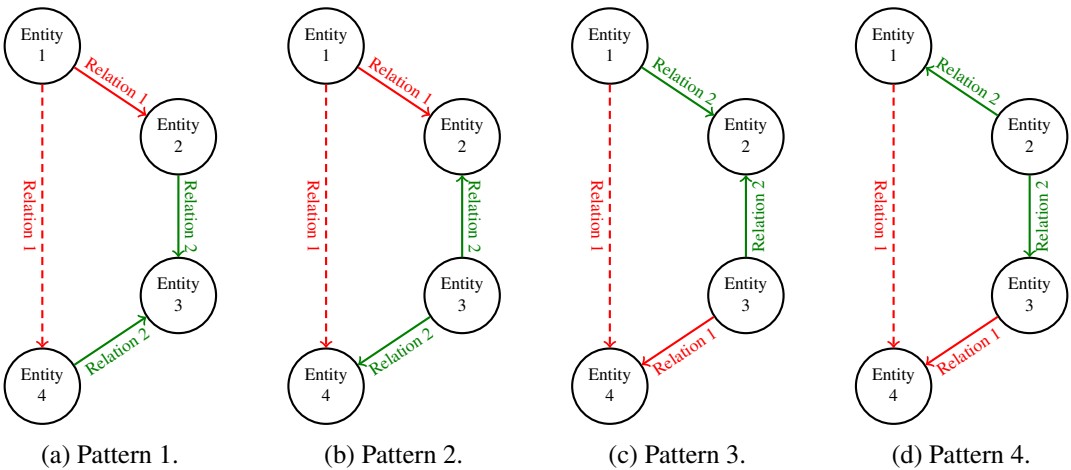

(a) Pattern 1.  (b) Pattern 2.  (c) Pattern 3.  (d) Pattern 4.

Figure 5: Simple patterns with length 3 which we incorporate to represent the WN18RR and YAGO3-10.

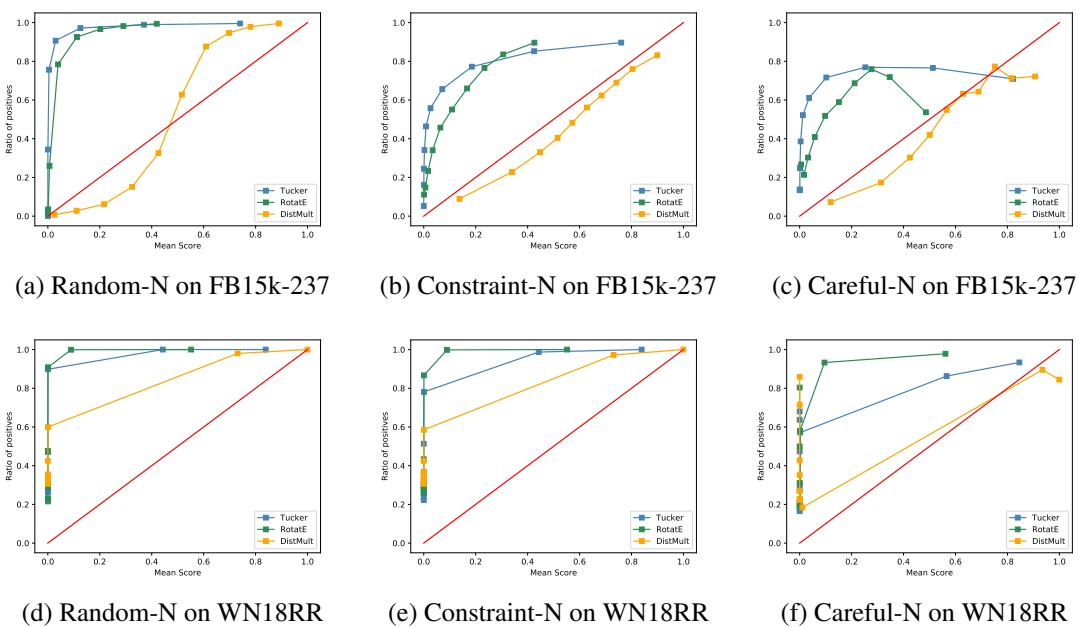

(a) Random-N on FB15k-237  (b) Constraint-N on FB15k-237  (c) Careful-N on FB15k-237

(d) Random-N on WN18RR  (e) Constraint-N on WN18RR  (f) Careful-N on WN18RR

Figure 6: Calibration study on different KGs based on three negative sampling procedures.

behavior is due to the fact that the link prediction models' performance tends to get saturated upon increasing the hidden dimension value.

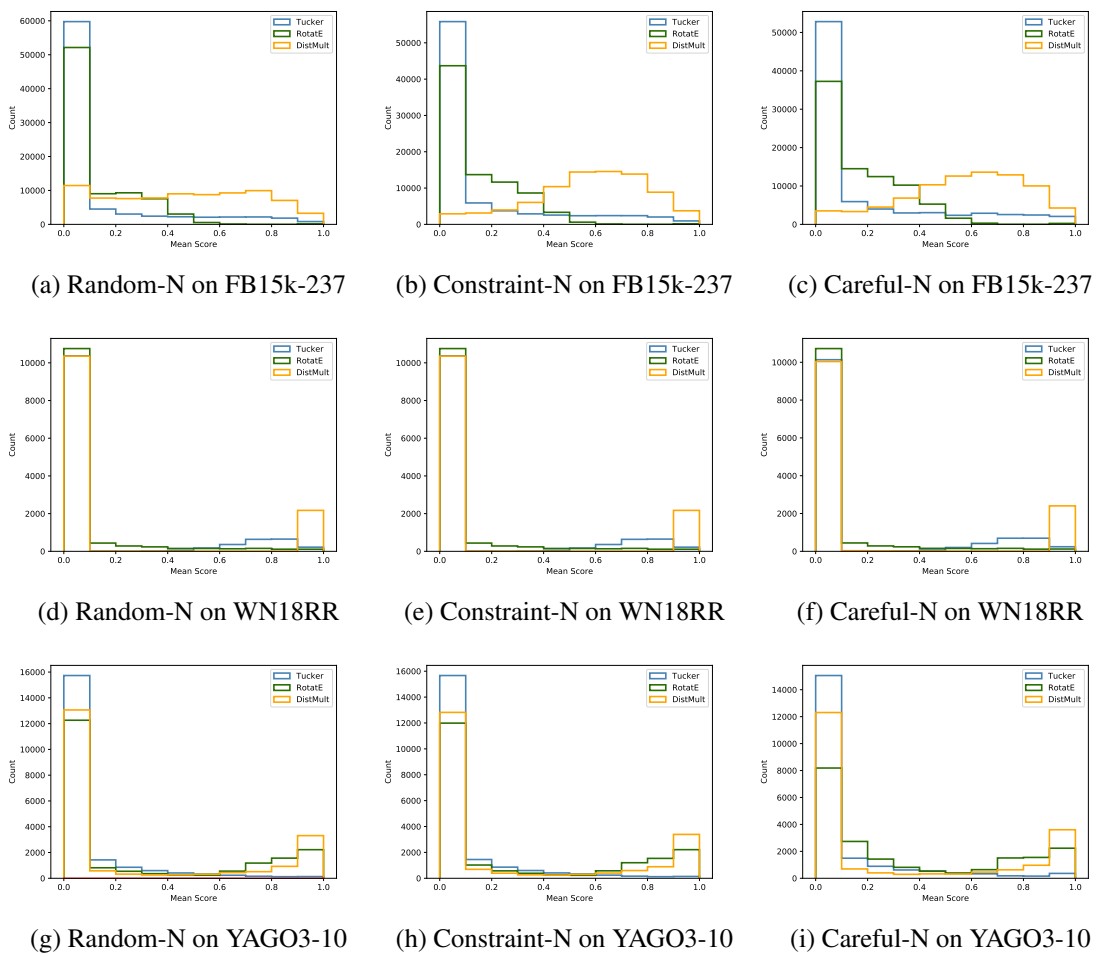

(a) Random-N on FB15k-237     (b) Constraint-N on FB15k-237     (c) Careful-N on FB15k-237

(d) Random-N on WN18RR     (e) Constraint-N on WN18RR     (f) Careful-N on WN18RR

(g) Random-N on YAGO3-10     (h) Constraint-N on YAGO3-10     (i) Careful-N on YAGO3-10

Figure 7: Calibration study on different KGs based on three negative sampling procedures.

Table 5: **Link Prediction** result for FB15k-237, WN18RR and YAGO3-10 KGs. All results generated by restricting the number of parameters to be equal to the DistMult's parameters with dimension 200.

| Models | FB15k-237 | | WN18RR | | YAGO3-10 | |
|---|---|---|---|---|---|---|
| | MRR | Hits@1 | MRR | Hits@1 | MRR | Hits@1 |
| DistMult | 0.279 | 17.9 | 0.39 | 36.4 | 0.423 | 33.8 |
| RotatE | 0.3 | 20.9 | 0.434 | 40.7 | 0.459 | 36.5 |
| Tucker | 0.339 | 25 | 0.423 | 40.4 | 0.417 | 33.4 |

## Appendix F. YAGO3-TC Relation Distribution

The relation distribution of YAGO3-10 test data on our randomly 1000 random sampled is depicted in Figure 10. As shows, except for relation *affiliatedTo* (relation 16) which we didn't consider in our sampling, other relations demonstrate similar distribution.

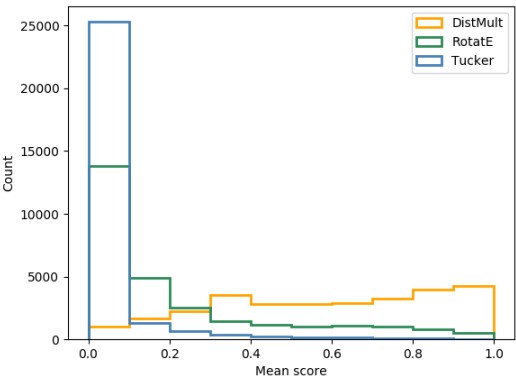

Figure 8: Histogram plot of Calibration on YAGO3-TC.

Table 6: **Per-Relation Breakdown**

| Relation | DistMult | | | | RotatE | | | | Tucker | | | |
|---|---|---|---|---|---|---|---|---|---|---|---|---|
| | Acc | F1 | R | P | Acc | F1 | R | P | Acc | F1 | R | P |
| playsFor | 25.9 | 23.2 | 85.6 | 13.4 | 20.6 | 22.8 | 89.8 | 13 | 73.5 | 29.1 | 41.6 | 22.4 |
| isLocatedIn | 35.4 | 23.2 | 83.8 | 13.5 | 21.7 | 20.3 | 85.6 | 11.5 | 45 | 23.7 | 73.4 | 14.1 |
| wasBornIn | 22.9 | 5.5 | 75.3 | 2.8 | 15.3 | 5.6 | 84.3 | 2.9 | 62.4 | 3.6 | 23.4 | 1.9 |
| hasGender | 78.4 | 32.4 | 92.6 | 19.7 | 94.7 | 45.3 | 38.9 | 54.4 | 97.9 | 82.2 | 85.2 | 79.4 |

## Appendix G. Per-Relation Breakdown

We perform a per-relation breakdown analysis on the YAGO3-TC dataset to gain a deeper understanding of how is the distribution of the model's performance on different relations. This kind of analysis can help us with identifying the shortcoming and the strength of our embedding methods. Table 6 compares RotatE and Tucker on the top four most frequent relations. As shown, RotatE outperforms Tucker in recall except for relation hasGender, and loses except for F1 and precision for relation wasBornIn. Relations playsFor and isLocatedIn show similar performance over all metrics in RotatE (and almost Tucker), demonstrating that these models learn similar pattern for these relations. Moreover, both models perform very poorly in relation wasBornIn, suggesting the difficulty in predicting this type of relation. While both models predict the relation hasGender with much more confidence, emphasizing the simplicity in the prediction of this relation.

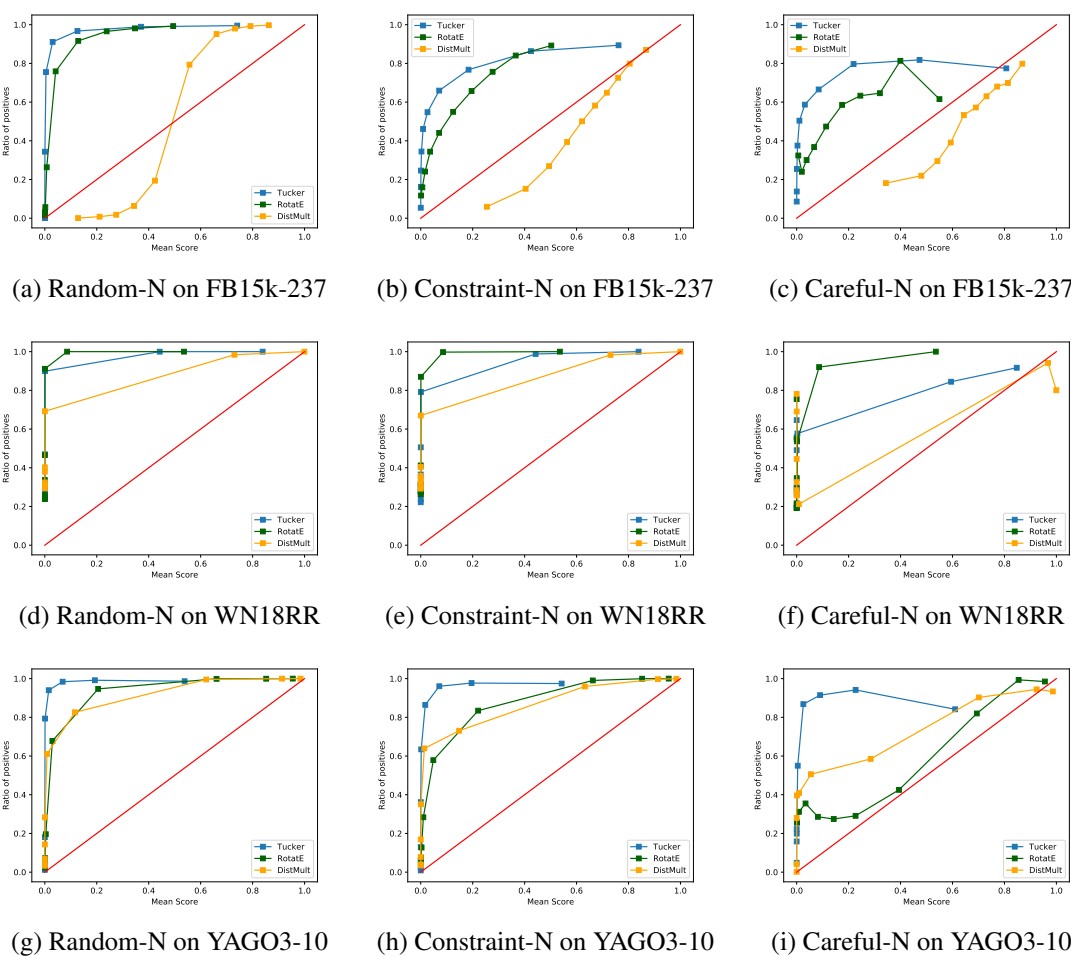

(a) Random-N on FB15k-237    (b) Constraint-N on FB15k-237    (c) Careful-N on FB15k-237

(d) Random-N on WN18RR    (e) Constraint-N on WN18RR    (f) Careful-N on WN18RR

(g) Random-N on YAGO3-10    (h) Constraint-N on YAGO3-10    (i) Careful-N on YAGO3-10

Figure 9: Calibration study on different KGs based on three negative sampling procedures.

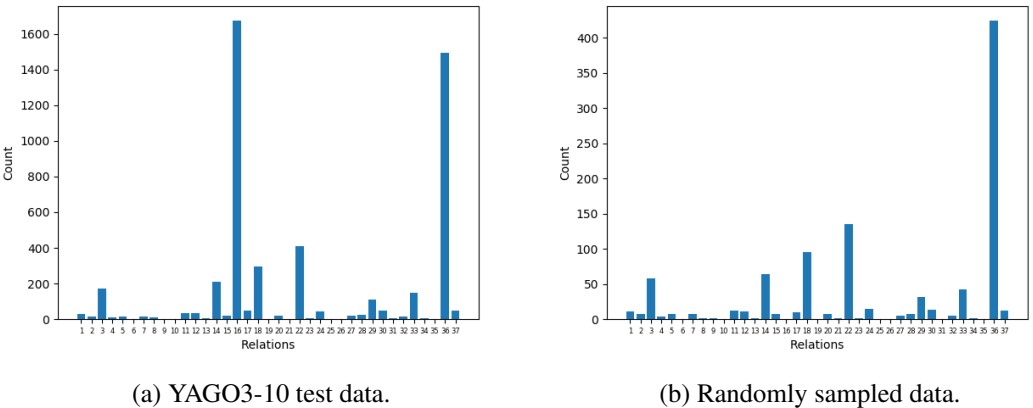

(a) YAGO3-10 test data.          (b) Randomly sampled data.

Figure 10: Distribution of relations in YAGO3-10 and our randomly 1000 sampled. Except for relation *affiliatedTo* (relation 16) which we didn't consider in our sampling, other relations demonstrate similar distribution.