# OpenReview forum: "Revisiting Evaluation of Knowledge Base Completion Models"
_AKBC.ws/2020/Conference — AKBC 2020_

### Official Review · AnonReviewer3 · 2020-03-10
**KGC evaluation analysis and classification dataset; experiments need improving**

**Rating:** 4
**Confidence:** 5

**Review:**

Summary: This paper studies the drawbacks of existing KGC evaluation methods, i.e. those based on ranking metrics and triple classification accuracy with negative samples. The authors propose a new KGC evaluation dataset (subset of YAGO3-10) and conduct extensive experiments on existing KGC models (DistMult, RotatE and TuckER).

The authors provide a much needed analysis into drawbacks of ranking metrics for KGC and stress the importance of classification-based evaluation. I appreciate the authors' claim that they will realease the code and datasets upon acceptance. However, I believe the paper in its current state is not ready for publication. Questions and more detailed comments below:

Section 3.1
1. Why do you think the existence of "semi-inverse" relations is a problem? This only occurs for symmetric relations (e.g. verb_group, also_see) and without these triples, I'm not sure how any model would learn that a given relation is symmetric.
2. How do you get the "0.95 MRR" figure? Does that mean that the MRR on all other (asymmetric) relations is very low? This is possibly due to WN18RR being very unbalanced, i.e. 2 relations dominate the dataset (derivationally_related_form and hypernym, see Table 3 in [1]) and the low MRR on asymmetric relations may be due to the low performance on the hypernym relation. It would be interesting to see per relation analysis of MRR for both datasets, similar to the one in [1].

Section 3.2
1. Are the plots in Figure 1 obtained with the same parameter numbers for each model as in Table 1? If so, I'm not sure these numbers are comparable, as RotatE for e.g. FB15k-237 has 5x more parameters than DistMult and 3x more than TuckER.
2. You don't define what the axes labels "Ratio of positives" and "Mean score" are in Figure 1.
3. It is not clear to me what the main takeaway from this section is, given that the results differ so much for different negative sampling procedures.
4. Do you have any explanation for the behaviour of TuckER in Figure 1c?

Section 3.3
1. I agree that the performance of rule-based and local approaches is higher than expected, but still, it is far from the performance of state-of-the-art models. To make the comparison fair, these approaches should be compared to the state-of-the-art models with a comparable number of parameters.

Section 3.4
1. Why is the relative ranking of models so different for Random-N vs Careful-N in Table 2? TuckER seems to be the worst performing model on easier Random-N, but perform the best on harder Careful-N.
2. Are different types of negative sampling applied only at test time, or also at training time?

Section 5
1. Why would you train on validation data? And how come this changes the relative ordering of performances across models? Is it because RotatE has a lot of parameters and it's esentially overfitting until you increase the amount of data?

Paper should be checked for word repetitions and typos. Writing quality could be improved.


[1] Balazevic et al. On Understanding Knowledge Graph Representation, https://openreview.net/forum?id=SygcSlHFvS&noteId=SygcSlHFvS.

---

> ### Author Response · Authors · 2020-04-08
> **Response to Reviewer #3**
>
> We would like to thank the reviewer for their helpful and detailed comments. We sincerely hope R3 can revisit the rating in light of our revision and response.
>
> “Issue with semi-inverse relations”
> We are not suggesting that semi-inverse relations are problematic for link prediction models, but they can be misleading for evaluation metrics. Since these relations consist of a huge part of test data, the performance of different link prediction models on the test data cannot capture the actual reseaning power of each model. Further, as we discussed at the end of section 3.1, since most of these relations are 1-N (N-1), and this N can be very large for some relations, the ranking metrics cannot capture unlabeled true facts properly for those relations.
>
> “MRR of semi-inverse relations”
> The reported 0.95 MRR performance, calculated assuming the test data only consists of the triples that have mentioned semi-inverse relations and their inverse appears in the training data. As the reviewer suggested, the MRR performance on other relations are much lower, which partly is because of the unbalanced nature of these KGs.
>
> “Number of parameters for different models in figure 1”
> Yes, we train each model with the best configuration provided in their related paper. Our goal was to only study the calibration on each model while having their best possible performance.
>
> “Labels of figure 1 axis”
> In our calibration figures, "Ratio of positives" denotes the fraction of number of positive triples to all of triples (for a specific score) and "Mean score" represents the score of each link prediction model for a target triple (normalized to 0-1 to correspond to confidence in ML studies). Let us note that, This representation is consistent with previous studies [1]. We define these labels more clearly in the revised version.
>
> “Take away from calibration section”
> In the calibration study, our goal was not only to show that these models may not be calibrated, but also that the calibration study itself is not well defined as a result of ambiguity in the possible different negative sampling procedures.
>
> “Explanation for Tucker behavior in figure 1c”
> We suspect the reason behind this particular performance is due to the fact that Tucker tends to score many triples (both positive and negative triples) very highly for some specific relations such as “hasGender” and “isLocatedIn”. We provide a more detailed explanation in the revised version.
>
> “Compare with models that have similar number of parameters “
> Since there are many entities in these KGs, even with the embedding size of 1, the SOTA models will have a few times more parameters compared to our models. Our goal in comparing these approaches with SOTA models was to show that a substantial portion of these models’ performance is coming from some very simple patterns, demonstrating that the high values of ranking metrics are not completely reliable.
>
> “Different relative ranking behavior for Random-N vs Careful-N”
> We believe that the reason behind different relative ranking behavior is the fact that since tucker scores many negative samples very highly (for some relations), it is harder for the classifier to distinguish between the positive and negative triples in Random-N scenario. Further, it seems Tucker scores positive and hard negative samples better in a way that make them more distinguishable in comparison to other models. We provide a more detailed explanation in the revised version.
>
> “Different negative sampling approach for test vs training”
> Since our goal was to evaluate the models fairly and under a unified setup, independent of their training strategy, we only apply the different negative sample schemas at the inference time.
>
> “Train on the validation data”
> We only train our classifier (tune the thresholds) on the collected validation data, which is widely adopted for triple classification tasks. Let us note that we train the classifier for each model after learning the link prediction task, and calculating scores for train (validation data in this case) and test data. As a result, these thresholds are parameters of the classifier instead of being part of the link prediction model necessarily. Our goal here was to show that we can potentially improve the performance of the classifier even by only using a small set of carefully chosen negative samples.
>
>
> [1] Tabacof, Pedro, and Luca Costabello. "Probability Calibration for Knowledge Graph Embedding Models." arXiv preprint arXiv:1912.10000 (2019).

---

> > ### Comment · AnonReviewer3 · 2020-04-13
> > **Rebuttal response**
> >
> > Thanks to the authors for their detailed response.
> >
> > Even though I believe the paper has merit (with the new classification dataset being its most important contribution), I stand by my decision that it is not ready for publication in its current state. Comparing existing models with orders of magnitude differences in numbers of parameters invalidates any conclusions of the proposed analysis.

---

> > > ### Author Response · Authors · 2020-04-16
> > > **Response to Reviewer #3**
> > >
> > > Thank you for your review and detailed feedback.
> > >
> > > In this work, we are interested in evaluating/analyzing the published models and existing research, which focus on optimizing accuracy/ranking metrics, without much regard for using similar model sizes. This is not a completely unreasonable perspective: larger models don't necessarily result in more accurate models (on this task), and achieving the highest performance on an important metric, while considering the number of parameters as a hyperparameter, is a useful endeavor. In this paper, we are questioning whether the popular metrics are in fact ones we should be relying on. Moreover, our specific goal in the calibration study is not to provide a canonical comparison of these models for calibration, but to show that calibration study itself is not well-defined and can lead to grossly inconsistent conclusions in the current evaluation system, which is a very different question than "is it fair to compare models with different number of parameters?".
> > >
> > > That said, we added results by using similar number of parameters over different models in the Appendix E. We consider the DistMult’s number of parameters with a hidden dimension of 200 as our benchmark. The results appear very similar to previously reported ones. The reason behind similar behavior is due to the fact that the link prediction models’ performance tends to get saturated upon increasing the hidden dimension value.

---

### Official Review · AnonReviewer1 · 2020-03-18
**A good analysis on popular KB-completion datasets plus a carefully labeled triple classification dataset**

**Rating:** 7
**Confidence:** 4

**Review:**

This paper first analyzes several popular KB-completion datasets and their evaluation methods.  Several issues have been highlighted and discussed, such as the assumptions made in the ranking metrics, skewed distributions on semi-inverse relations (in WN18RR & YAGO3-10), confidence scores by popular methods are not calibrated.  In addition, the authors also suggest that some simple baselines are actually quite robust.  Based on their finding, the author creates a binary triple classification dataset.  Effectively, every triples in their dataset are examined by multiple Turkers to ensure the label quality and also to avoid the potential error due to the "close-world" assumption behind some existing datasets.

General comments:

I'm happy to see that the authors revisit the problem of how KB-completion is evaluated.  Although the various potential issues of existing datasets and/or evaluation metrics are not necessarily secrete to KB-completion researchers, it is still good to identify and discuss them.  While I agree most of the analysis and findings, I have to argue that the reason behind those issues is often that the use case was not carefully discussed and defined first.  As a result, it is very easy to find special biases or skewed distributions of some triples, which may be exploited by different models.

The proposed YAGO3-TC dataset, in my opinion, is one step towards to right direction.  Setting it up as a simple binary classification problem of whether a triple is correct, avoids the implicit and incorrect "close-world" assumption, and thus ensures the label correctness.  The task is more like fact-checking or simple question answering.  However, the potential issue of this dataset is the distribution of the triples.  Because it is somewhat selected by two existing methods, it could be sub-optimal compared to, say, triples generated by some human users with a specific scenario in mind.

Detailed comments:

	1. It is a common and well-known issue that the probabilities or confidence scores of ML models are not calibrated.  It is not surprising to see that this problem also exists in KB-completion models.  However, given that dev sets are available, why didn't the author apply existing calibration methods (e.g., those mentioned in Guo et al., ICML-17) to the output of the existing models?
	2. Similarly, the type information can be used in conjunction with the existing models, even as a post-processing step (e.g., see [1]).  The performance of existing models may be improved substantially.
	3. For imbalanced class distribution, the "accuracy" metric is not very meaningful.  Precision/Recall/F1 are better.  Another alternative is the ROC analysis (False-positive rate vs. True-positive rate) if the task can be cast as an anomaly detection problem.  Again, the choice of evaluation metrics depends on the underlying use-case scenario.
	4. Probably too many details and discussions are put in Appendix.

[1] Chang et al., Typed Tensor Decomposition of Knowledge Bases for Relation Extraction. EMNLP-2014.

---

> ### Author Response · Authors · 2020-04-08
> **Response to Reviewer #2**
>
> We would like to thank the reviewer for their helpful and detailed comments, and have attempted to address the concerns.
>
> “Applying existing calibration methods”
> In the calibration study, our goal was to show that these models are not only uncalibrated, but also that the calibration study itself is not well defined as a result of ambiguity in the negative sampling procedure. Further, since calibration looks very different based on different sampling techniques, it is not clear how to use those methods to improve link prediction model’ calibration.
>
> “Incorporating type information”
> Although incorporating the type information can improve the performance of link prediction, since we do not have access to this kind of information for many KGs, it is not considered to be part of the problem setup of KG completion, and arguably the models are supposed to be able to infer them from the relations.
>
> “ROC metric”
> Good idea! We added the ROC AUC result of the classifiers on YAGO-TC data. As you can see the results are very similar to F1 behavior for different models.
>
> “Too many details in appendix”
> We will revisit the appendix in the light of identifying the pieces to be removed, and added to the main text. We really appreciate it if the reviewer can suggest any pieces of information that he/she finds necessary to be added into the main text.

---

### Official Review · AnonReviewer2 · 2020-03-30
**A timely and thorough analysis of evaluation of knowledge base completion and a new evaluation benchmark**

**Rating:** 9
**Confidence:** 4

**Review:**

This paper conducts a systematic and thorough analysis of a long-criticized problem of current knowledge base completion (KBC) research -- its evaluation setup. It discovers several major issues of the popular ranking metric with negative sampling and several popular benchmarks, such as unrealistic assumptions, existence of semi-inverse relationships in the KBs, suspicious model calibration behavior, and all together leading to inconclusive evaluation of true model performance. Based on the analysis, the paper further proposes that triple classification is a better metric for KBC because it's less prone to the problems from the open-world nature of KBs and their incompleteness. However, triple classification doesn't work well with randomly sampled negative examples (as shown in the paper). Therefore, this paper also collects a new benchmark from crowdsourcing based on YAGO3-10 where the positive and negative triples are examined and judged by multiple workers, hence forming a more solid ground for evaluation than random sampling. Several simple heuristic baselines are also proposed and shown to perform comparatively with state-of-the-art embedding models on the new benchmark, which shows that there's still a large room for model development.

Strengthes:

- This is a timely and thorough analytical study on a lone-criticized issue of a popular and important research problem

- The investigations are well-designed and show many interesting insights

- The new benchmark is also a very good contribution to the field and likely will be used by many studies in the future

- Overall the paper is very well written and easy to follow

Weaknesses

- If we think about the ways KBC methods may be used in practical applications, ranking metrics do have their use cases and triple classification may not be the only (or strictly better) metric. For some applications we may have specific hypotheses to validate, for which triple classification may be better. But for some other, more exploratory applications, e.g., discovering new chemical compounds for certain purpose, it may be actually preferred to produce a ranked list of hypotheses and validate the hypotheses using other experiments. Personally I think negative sampling is the true problem of ranking metric due to the incompleteness of KBs, not the ranking nature itself.

- It seems that only minimal quality control of crowdsourcing was implemented. Crowd workers could make mistakes, and different workers could produce work of very different quality. The described setup seems to be prone to worker annotation errors. It is mentioned that "to check the quality of our labels, we randomly consider 100 triples from test data in our study. As a result, 96% of these triples considered to be positive by users in our study, demonstrating the high quality of our labels" . How was this check done?

- The validation set is quite small. Not sure whether it's sufficient to ensure robust modeling decisions.

- The paper could benefit from more fine-grained analysis on the new benchmark and possibly pointing out promising venues for future improvement.

---

> ### Author Response · Authors · 2020-04-08
> **Response to Reviewer #1**
>
> We would like to thank the reviewer for their helpful and detailed comments. We are glad that the reviewer finds our paper interesting.
>
> “Practical usage of ranking metrics”
> We agree with the reviewer on the potential real-world application of ranking metrics. In this work, we are not suggesting completely replacing ranking for all use cases, but pointing out their shortcomings, and introducing a benchmark to compute other metrics. We made this clear in the revised version of the paper.
>
> “Quality evaluation of crowd workers results”
> To check the quality of annotations, we randomly put 100 of the test triples (which are true) as possible extra choices for crowd workers to choose from. Then, applying our voting schema, we observe that 96% of these triples, considered to be positive by crowd workers as well. To further improve the quality of our labels, we first identify the annotated labels with a higher possibility of mistake using link prediction models’ score (very high score triples that are labeled as negative and very low score ones that labeled as positive). As a result, we end up with around 200 labels which we then manually check their correctness ourselves.
>
> “Size of validation set”
> We agree that the small size of validation data might affect the robustness of the classifier. Let us note that, our validation data’s size is roughly a third of the training data used commonly for training triple classification task. Further, considering the expensive nature of data collection and our effort to provide a well-distributed validation data (over relations), we believe our validation data should provide a sufficient source of samples to train a robust triple classifier.

---

### Decision · Program_Chairs · 2020-04-30

**Decision:**

Accept

**Comment:**

This paper points issues with evaluation of knowledge base completion, proposes triple classification as an evaluation method, and introduces a new dataset to help with this. In this sense, the paper is essentially an analysis paper that focuses on the evaluation aspects of the problem.

All the reviewers appreciate the analysis of evaluation metrics for KG completion and also the new dataset. The authors have updated the paper to address many of the concerns raised by in the reviews, in many cases providing additional information to make their case.